# Referring Multiple Regions with Large Multimodal Models via Contextual Latent Steering

**Yun Xing** [1]  **Hanyuan Liu** [1]  **Jiahao Nie** [1]  **Shijian Lu** [1]

## Abstract

Large Multimodal Models (LMMs) have recently demonstrated their proficiency in holistic visual comprehension. However, most of them struggle to tackle region-level perception guided by visual prompts, especially for cases where multiple regions are referred simultaneously, or scenarios where global contexts are necessary for precise visual referring. We introduce **Contextual Latent Steering (CSteer)**, a training-free approach for guiding general LMMs to refer multiple regions contextually, without expensive fine-tuning or architectural modifications. **CSteer** starts with pre-computing *contextual vectors* that implicitly represent visual referring behaviors, such as differentiation among regions and attention to global contexts, followed by representation editing during inference time. Experimental results on multiple datasets indicate that general LMMs with **CSteer** outperform tailored referring LMMs in most cases, suggesting a promising solution in training-free, and setting new state-of-the-art for this field. Code is available at `https://github.com/xing0047/csteer.git`.

## 1. Introduction

Large Multimodal Models (LMMs) (Liu et al., 2023; Dai et al., 2023; Bai et al., 2025; Wang et al., 2025c) have demonstrated their proficiency in general image and video understanding, such as visual question answering (Li et al., 2023; 2024b) or describing scenes in detail (Chai et al., 2024; Maaz et al., 2024). Although existing LMMs excel in holistic visual understanding, most of them struggle to perceive and reason over localized regions, known as *visual referring* (Wang et al., 2025b; Lian et al., 2025). This typically requires precise interpretation at the instance-level (Peng

[1]Nanyang Technological University. Correspondence to: Shijian Lu <shijian.lu@ntu.edu.sg>.

*Proceedings of the 43rd International Conference on Machine Learning*, Seoul, South Korea. PMLR 306, 2026. Copyright 2026 by the author(s).

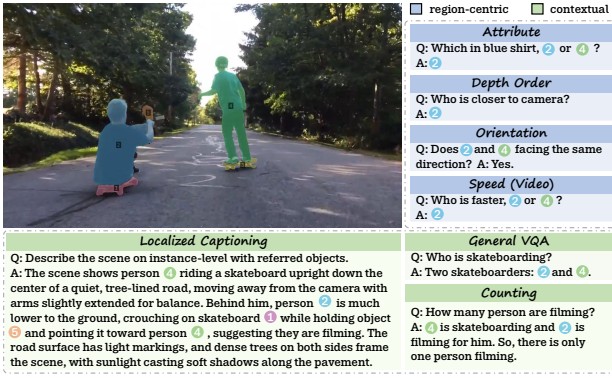

*Figure 1.* An example from (Peng et al., 2024) demonstrating *multi-region visual referring*, with a wide spectrum of region-focused scenarios and instance-level cognition.

et al., 2024), thus powering a wide range of applications with more nuanced granularity, such as robotics (Yuan et al., 2024a) or remote sensing (Zhang et al., 2024b).

There have been ongoing explorations in fine-tuning LMMs that excel at *visual referring* (You et al., 2023; Zhang et al., 2025b; Yuan et al., 2024b; Cai et al., 2024; Rasheed et al., 2024; Yuan et al., 2025c), which are recently capable of describing nuanced details for small targets (Lian et al., 2025), or capturing temporal dynamics for objects in videos (Yuan et al., 2025b). A common practice resorts to customized designs, such as employing another encoder for explicit tokenization of input regions (Rasheed et al., 2024; Zhang et al., 2024a; Yuan et al., 2025a; Lian et al., 2025; Yuan et al., 2025b), together with tailored fine-tuning data (Lian et al., 2025) or post-training recipe (Wang et al., 2025b).

Although these studies bring advances to *visual referring*, most of them focus on perceiving one isolated region, while *multi-region visual referring* is less explored (Wang et al., 2025b; Peng et al., 2024). This examines how LMMs misperceive when multiple regions are referred simultaneously. *Multi-region visual referring* naturally involves two key aspects, as given in Fig. 1. One is more *region-centric*, usually involving comparisons among regions, such as assessing orientations and depths (Fu et al., 2024; Tong et al., 2024). Another is more *contextual*, which requires precise global understanding before referring accurately, such as counting or localized captioning (Peng et al., 2024), as in Fig. 1.

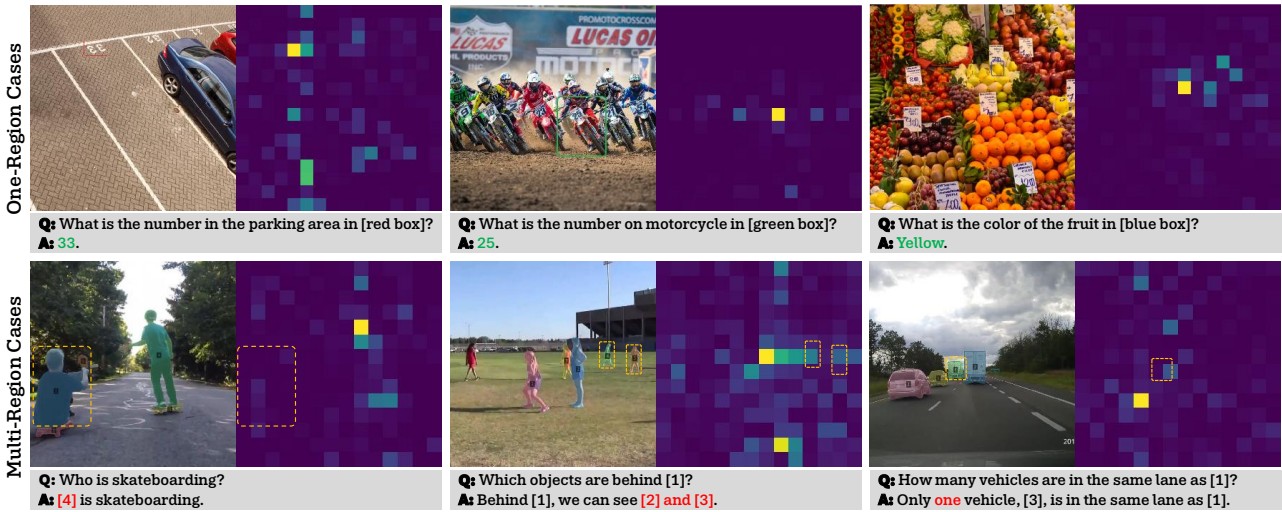

*Figure 2.* LMMs know where to look when referring one region (Zhang et al., 2025a) but struggle when multiple regions are visually referred. We use Qwen3-VL-8B (Bai et al., 2025) for inference of the examples, where orange means regions not referred.

Instead of introducing customized designs for *multi-region visual referring* and expensive task-oriented instruction tuning, we explore how to trigger such capability from general LMMs in a *training-free* manner (Yang et al., 2023). General LMMs like Qwen3-VL (Bai et al., 2025) are able to perceive visual marks naturally, such as points, boxes, and masks with numerical identifiers. Moreover, these LMMs know where to look (Zhang et al., 2025a) with simple one-region referring cases, as in Fig. 2 (top). Despite its simplicity, Set-of-Mark (Yang et al., 2023) over general LMMs is not necessarily effective for multi-region cases, especially for *contextual* ones that require precise global understanding (*e.g.*, Fig. 2 bottom, where attentions (Zhang et al., 2025a) are not comparably faithful like one-region cases).

Motivated by (Rimsky et al., 2024), we propose **Contextual Latent Steering (CSteer)**, a training-free solution that triggers *multi-region visual referring* directly from general LMMs. **CSteer** pre-computes *contextual vectors* that implicitly encode behaviors of *multi-region visual referring*, such as differentiation among regions and more attention to global contexts, followed by applying them to the latent representations during inference. *Contextual vectors* capture underlying changes when a LLM judge rewrites erroneous localized captions from general LMMs, such as rectifying mismatched numerical identifiers or complementing missing regions (*e.g.*, Fig. 2). Empirically, **CSteer** on the most recent general LMMs (*e.g.*, Qwen3-VL (Bai et al., 2025)) outperforms state-of-the-art tailored referring LMMs (Peng et al., 2024; Lian et al., 2025) on multiple representative benchmarks (Wang et al., 2025b; Peng et al., 2024; Fu et al., 2024), including the recent ones that require precise contextual referring (Wang et al., 2025b), indicating its strong potential in addressing multi-region scenarios.

We summarize our contributions as follows,

- We propose Contextual Latent Steering (CSteer), a simple yet effective training-free approach that enables general LMMs to visually refer with multiple interactive prompts simultaneously, without tailored architecture, expensive supervised fine-tuning or policy optimization.

- We perform thorough ablations, suggesting multiple design choices for both generating contextual vectors and steering. We emphasize our approach by vectoring with referential rewrites and steering with decomposition.

- On top of the recent general LMMs, CSteer achieves the state-of-the-art over multiple visual referring benchmarks, such as GAR-Bench and INST-IT characterized by multi-region and contextual referring.

## 2. Related Works

**Large Multimodal Models**. Recently, Large Multimodal Models (LMMs) (Liu et al., 2023; Bai et al., 2025; Dai et al., 2023; Wang et al., 2025c; Chen et al., 2023; Li et al., 2024a; Guo et al., 2025) have achieved remarkable milestones, enabling applications such as agents (Xie et al., 2024; Zheng et al., 2025b) or embodied intelligence (Kim et al., 2024; Zhou et al., 2025). Early LMM works (Liu et al., 2023; Dai et al., 2023) address perceptual tasks (Chen et al., 2024a; Liu et al., 2024; Li et al., 2024b), while recently more studies (Wang et al., 2025a; Shu et al., 2025) focus on tackling complex reasoning scenarios (Xiao et al., 2024; Cheng et al., 2025). Although existing LMMs excel in the global understanding of images and videos (Zhang et al., 2023; Nie et al., 2026), they struggle to describe and reason at a finer-grained region-level (Lian et al., 2025; Yuan et al., 2025c).

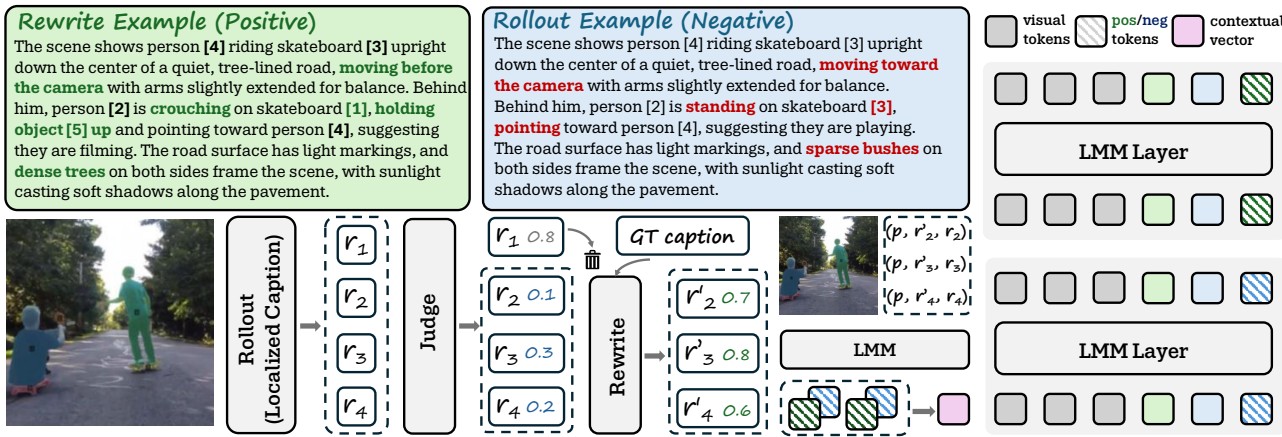

*Figure 3.* **Contextual Vector**. (*left*) CSteer first obtains incorrect localized captions and corrected referential rewrites with the aid of a text-only judge, and ground-truth captions as reference. (*right*) The incorrect captions and their paired rewrites then derive positive and negative last tokens via teacher forcing (pos/neg tokens), which are contrasted and averaged across data for building contextual vectors.

**Visual Referring**. *Visual referring* has long been developed to empower LMMs with region-level cognition capabilities (Chen et al., 2023; You et al., 2023; Yuan et al., 2024b; Zhang et al., 2025c; Rasheed et al., 2024; Zhang et al., 2024a; Lin et al., 2024; Lian et al., 2025; Yuan et al., 2025a; Sun et al., 2025). Specifically, they often involve an additional region encoding module to obtain local region-centric features (Zhang et al., 2025c; Rasheed et al., 2024; Yuan et al., 2025c; Lian et al., 2025), with a large collection of region-referred data used for instruction tuning (Yuan et al., 2024b) or reinforcement post-training (Wang et al., 2025b). Some studies (Cai et al., 2024; Peng et al., 2024) train LMMs to recognize visual prompts (Yang et al., 2023) with customized referential dialogs instead. Most existing referring LMMs are proficient in single object captioning (Lian et al., 2025) but fail when handling multi-region cases. Our approach guides LMMs to recognize not only isolated, but also multiple regions with visual prompts by training-free steering (Rimsky et al., 2024; Arditi et al., 2024; Wu et al., 2025b; Stolfo et al., 2024; Chalnev et al., 2024).

**Multi-Region Understanding**. Although existing LMMs excel at perceiving isolated objects, an arbitrary number of instances or visual prompts at the input-level could pose a challenge to these models (Wang et al., 2025b; Peng et al., 2024). This typically includes spatial or comparative relations (Wang et al., 2024; Guan et al., 2024; Li et al., 2023; Nie et al., 2024; Zheng et al., 2025a), object counting (Paiss et al., 2023; Arteta et al., 2014; Amini-Naieni et al., 2024; Amini-Naieni & Zisserman, 2025), hallucinations (Chen et al., 2024b), and contextual referring (Cai et al., 2024; Wang et al., 2025b). This could be further demanding if videos are input, considering the disappearance or reappearance of instances (Seo et al., 2020; Ravi et al., 2024; Ding et al., 2025). In our paper, we study how general LMMs without tailored fine-tuning behave when multi-region sce-

narios are given, and how we improve such understanding in a training-free manner.

## 3. CSteer

In this section, we introduce our **Contextual Latent Steering (CSteer)**, a training-free approach that guides general LMMs to refer multiple regions contextually without tailored fine-tuning. As preliminaries, we first briefly describe how general LMMs tackle visual input, regardless of images and videos in Sec. 3.1. Then, we present two consecutive modules in **CSteer**, including how *contextual vectors* are generated in Sec. 3.2, and how these vectors are leveraged for intervening LMM inference in Sec. 3.3. For both modules, we consider several comparable alternatives for detailed ablations, highlighting designs that are crucial for referring multiple regions contextually.

### 3.1. Overview

**LMM**. Given a visual input $v$ and query input $q$, LMM $\mathcal{F}$ would typically start with encoding $v$ as $\mathbf{X}_v$, with a pre-trained visual encoder $\mathcal{F}_v$ and a projection module $\mathcal{F}_{proj}$. The query input $q$ is processed by the LMM tokenizer $\mathcal{F}_{tok}$ as $\mathbf{X}_q$, which is concatenated with $\mathbf{X}_v$ as $\{\mathbf{X}_v, \mathbf{X}_q\}$ to construct the input $\mathbf{X}_c$ to the language model $\mathcal{F}_L$. At each decoding step $t$, $\mathcal{F}_L$ samples a new token $x_t$ conditioned on both $\mathbf{X}_c$ and previously generated tokens $\mathbf{X}_{<t}$.

$$P(x_t \mid \mathbf{X}_c, \mathbf{X}_{<t}) = softmax(\mathcal{H}(h_c^L, h_{<t}^L)) \qquad (1)$$

where $\mathcal{H}$ stands for the head projection of LLM and $h^L$ represents hidden states from the last layer of $\mathcal{F}_L$.

**Visual Referring**. To enable LMM $\mathcal{F}$ to perceive and reason at the region-level, additional visual localizations are given

as input, noted as $p$ from $\{p_1, ..., p_m\}$. For multimodal settings at the global-level, $m$ is degenerated to $0$. The specific form of $p$ can be points, boxes, masks, or numerical texts distinguishing instances (Peng et al., 2024) or video frames (Wu et al., 2025a), where the goal is to inject $p$ into LMMs. A common practice is to use a retrained encoder $\mathcal{F}_{region}$ and encode $p$ to tokens. In these cases, the input to the language model $\mathcal{F}_L$ can be simplified as

$$\mathbf{X}_c = \text{concat}(\mathcal{F}_{visual}(v), \mathcal{F}_{region}(p), \mathbf{X}_q) \quad (2)$$

This form is adopted by tailored training approaches (Yuan et al., 2025a; Lian et al., 2025; Wang et al., 2025b). Another type of design for visual referring is to directly overlay $p$ on visual input $v$ (Cai et al., 2024; Yang et al., 2023), where the input is formulated as

$$\mathbf{X}_c = \text{concat}(\mathcal{F}_{visual}(v, p), \mathbf{X}_q) \quad (3)$$

This alleviates the need for training an additional region encoder $\mathcal{F}_{region}$, which are more flexible for tackling diverse visual referring scenarios without involving additional training, such as interacting with points (Mani et al., 2020) beside boxes, or multi-region cases in our paper (Tong et al., 2024; Wang et al., 2025b).

### 3.2. Contextual Vector

Motivated by (Rimsky et al., 2024), we begin by building contrastive pairs for the generation of steering vectors. The key is to implicitly empower the vectors with referential capability.

**General Form**. Given visual input $v$, the steering vectors $\Delta$ are typically from an example set $\{v, p, x_+, x_-\}$. The vectors $\Delta$ are computed with

$$\Delta^l = h_+^l - h_-^l = f_+(v, p, x_+) - f_-(v, p, x_-) \quad (4)$$

where $x_+$ and $x_-$ represent expected and contrastive behaviors, respectively. Through $f$, activations are derived from positive $x_+$ or negative samples $x_-$, by forcing $\mathcal{F}$ to answer $x_+$ or $x_-$ respectively. Typically, the last activations are editing directions.

To generate referring vectors for multi-object cases, we include multiple design choices, such as

**Refer v.s. No Refer**. For this design choice, $f_+$ overlays visual prompts $p$ on the same input $v$, following (Yang et al., 2023), while $f_-$ ignores $p$. As given in Fig. 3, $x_+$ forces activations that refers correctly while $x_-$ leads to deliberate errors, namely

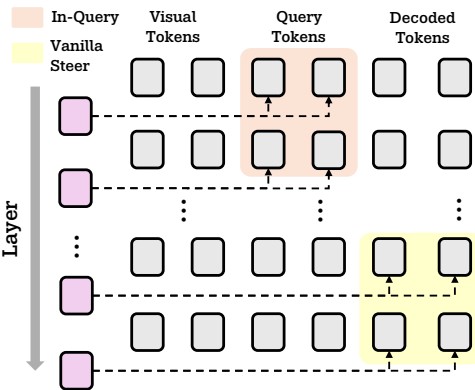

*Figure 4.* **Layer Decomposed Steering**. In CSteer, we apply *contextual vectors* in both queries at early layers and during decoding at middle-to-last layers.

$$\Delta^l = f_+(v, p, x_+) - f_-(v, x_-) \quad (5)$$

**Exact Matching v.s. Marker Shuffle**. The overlayed visual prompts should match exactly to the right regions. Unlike Eq. 5, intentional false matching could provide contrastive guidance as well via

$$\Delta^l = f_+(v, p, x_+) - f_-(v, p_{shuffled}, x_-) \quad (6)$$

**GT v.s. Rollout**. Drawing from inspirations that LMMs could correct themselves (Kumar et al., 2024), we hope to elicit visual referring capabilities of LMMs with corrective pairs. The sampled false rollouts are used for constructing referring vectors, while the correct ones are ignored, namely

$$\Delta^l = f_+(v, p, x_+) - f_-(v, p, \hat{x}_-) \quad (7)$$

where $\hat{x}$ represents LMM predictions instead of labels.

**Rewrite v.s. Rollout**. Another vectoring solution is from the perspective of positive samples, by rewriting incorrect LMM rollouts. This could be achieved with a text-only LLM $F_{rewrite}$ (*e.g.*, GPT-4o (Hurst et al., 2024)) to correct rollouts with ground-truth captions as references, namely

$$\Delta^l = f_+(v, p, \mathcal{F}_{rewrite}(\hat{x})_+) - f_-(v, p, \hat{x}_-) \quad (8)$$

### 3.3. Layer Decomposed Steering

During LMM inference, the previously constructed referring vectors are applied to all decoding steps by default

$$\hat{h}_t^l = h_t^l + \lambda \cdot \Delta^l \quad (9)$$

**Marker-Only**. Rather than applying visual referring vectors to all decoding steps, our approach focuses only on the

*Table 1.* GARBench (Wang et al., 2025b). SoM is short for Set-of-Mark prompting (Yang et al., 2023).

| Model | | MC | | | | | | | | OE | | |
| --- | --- | --- | --- | --- | --- | --- | --- | --- | --- | --- | --- | --- |
| | | CLR | SHP | TEX | MAT | POS | NET | REL | ALL | SIM | DET | ALL |
| Proprietary LMMs | | | | | | | | | | | | |
| GPT-4o | | 34.8 | 65.3 | 48.3 | 52.8 | 57.8 | 60.2 | 61.4 | 53.5 | 39.2 | 62.6 | 51.5 |
| o3 | | 58.0 | 70.3 | 55.2 | 63.9 | 54.7 | 49.2 | 71.3 | 61.3 | 37.1 | 74.8 | 56.9 |
| Gemini-2.5-Pro | | 62.3 | 68.8 | 58.6 | 66.7 | 64.1 | 64.9 | 70.3 | 64.2 | 51.6 | 66.4 | 59.3 |
| Region LMMs | | | | | | | | | | | | |
| Sa2VA 8B | | 39.1 | 45.3 | 29.6 | 30.6 | 54.7 | 21.3 | 21.8 | 34.3 | 46.4 | 44.9 | 45.6 |
| VP-SPHINX 13B | | 33.3 | 25.0 | 44.8 | 38.9 | 60.9 | 34.3 | 32.7 | 37.5 | 27.8 | 39.3 | 32.3 |
| GAR 8B | | 59.4 | 54.7 | 75.9 | 52.8 | 48.4 | 60.7 | 68.3 | 59.9 | 66.0 | 64.5 | 62.2 |
| General LMMs *w.* Training-Free Methods | | | | | | | | | | | | |
| InternVL-3.5 8B | *w/o* refer | 24.6 | 50.0 | 48.3 | 30.6 | 43.8 | 24.6 | 44.6 | 38.2 | 27.8 | 46.7 | 38.7 |
| | SoM | 43.5 | 50.0 | 51.7 | 44.4 | 67.2 | 34.4 | 58.4 | 50.9 | 27.8 | 49.5 | 39.2 |
| | CSteer | 44.9 | 53.1 | 51.7 | 47.2 | 67.2 | 37.7 | 61.4 | 53.1 | 29.9 | 55.1 | 43.1 |
| Qwen3-VL 8B | *w/o* refer | 31.9 | 43.7 | 34.5 | 44.4 | 45.3 | 26.2 | 45.5 | 39.3 | 24.7 | 39.2 | 31.8 |
| | SoM | 63.7 | 64.0 | 65.5 | 63.8 | 73.4 | 59.0 | 60.4 | 63.9 | 45.4 | 58.9 | 52.5 |
| | CSteer | 66.6 | 64.0 | 65.5 | 63.8 | 81.2 | 59.0 | 61.4 | 65.8 | 47.4 | 66.4 | 57.4 |

marker tokens $h_{t \in \mathcal{P}}^l$, where $\mathcal{P}$ is the set of marker tokens $\{p_1^t, ..., p_M^t\}$. We observe that steering only these tokens during inference has minimal impact on visual referring of LMMs (as in Tab. 5).

**In-Query**. For LLM behaviors such as sychophancy or hallucination (Rimsky et al., 2024), we notice that they only consider decoding steps, while for visual referring, marker tokens (*e.g.*, "[1]" in Fig. 3) are included in user queries. We also consider editing marker tokens in query, where experiments show that visual referring vectors are also applicable to a multimodal context $X_c$.

**Decomposition**. Driven by existing study in LLM information flow (Wang et al., 2023), we point out that the two design options mentioned above are complementary. In particular, the in-query option relates closely to information aggregation, while the marker only option is for the decoding stage that predicts outputs. In our results, we show that these two favor different parts of LMMs, with in-query favoring early LLM layers, and marker-only decoding favors middle to late LLM layers.

## 4. Experiments

### 4.1. Implementation Details

**Models**. We benchmark our design with various types of LMMs, including proprietary ones with multimodal understanding and reasoning capability, such as GPT-4o, o3, and Gemini-2.5-Pro (Hurst et al., 2024; OpenAI, 2025;

Comanici et al., 2025). We also benchmark with region LMMs that are trained with tailored referring data, including Sa2VA, DAM, INST-IT-Qwen and GAR (Yuan et al., 2025a; Lian et al., 2025; Wang et al., 2025b; Peng et al., 2024). Note that our design is plug-and-play to general LMMs, for which we use the most recent InternVL-3.5 and Qwen3-VL (Wang et al., 2025c; Bai et al., 2025), which are state-of-the-art in many understanding test scenarios.

**Benchmarks**. We use mainly two recent multi-region and contextual visual referring benchmarks. In particular, GAR-Bench (Wang et al., 2025b) is characterized by multi-region and contextual cases, with diverse referring aspects, including color, shape, or texture. INST-IT (Peng et al., 2024) is characterized by discriminations at the instance-level, with both images and videos as input domains, both multiple-choice and open-ended questions as prompts. We also include some other referring benchmarks, such as VIP-Bench (Cai et al., 2024) and a few subsets from BLINK (Fu et al., 2024) that target multi-region scenarios, such as comparing reflectance or depth.

### 4.2. Main Results

**GAR**. We first evaluate the recent GAR-Bench (Wang et al., 2025b) that examines multi-region scenarios, with considering contextual referring in their benchmarks. The results are given in Tab. 1. We point out that Qwen3-VL with a simple prompting approach (Yang et al., 2023) achieves the state-of-the-art in both multiple-choice (MC) and open-ended generation (OE) tasks, surpassing the recent state-of-the-art

*Table 2.* VIP (Cai et al., 2024) and BLINK (Fu et al., 2024). VIP measures one-region and contextual visual referring with diverse input domains, while BLINK measures comparing relations with visual markers. Subsets are shorted and details are mentioned in Appendix.

| Model | | REC | OCR | KNO | MAT | REL | LAN | AVG | RRF | RDP | FCO | ALL |
|---|---|---|---|---|---|---|---|---|---|---|---|---|
| | | | | | VIP | | | | | BLINK$_{val}$ | | |
| Proprietary LMMs | | | | | | | | | | | | |
| GPT-4o | | 56.5 | 63.6 | 56.8 | 70.3 | 53.9 | 51.2 | 58.7 | 76.9 | 56.0 | 38.8 | 57.2 |
| o3 | | 65.6 | 79.9 | 70.8 | 81.6 | 79.3 | 61.9 | 73.1 | 51.5 | 76.6 | 65.3 | 64.2 |
| Gemini-2.5-Pro | | 81.3 | 85.7 | 79.2 | 94.8 | 87.5 | 72.5 | 82.9 | 61.2 | 83.1 | 69.2 | 70.9 |
| Region LMMs | | | | | | | | | | | | |
| Inst-IT-Qwen 7B | | 58.9 | 24.5 | 48.5 | 12.9 | 48.2 | 46.3 | 39.9 | - | - | - | - |
| VIP-LLaVA 7B | | 56.3 | 24.6 | 53.4 | 15.5 | 50.0 | 53.8 | 42.3 | 26.8 | 51.6 | 21.5 | 33.0 |
| General LMMs *w.* Training-Free Methods | | | | | | | | | | | | |
| InternVL-3.5 8B | *w/o* refer | 51.2 | 45.7 | 56.1 | 32.9 | 56.1 | 61.3 | 49.6 | - | - | - | - |
| | SoM | 67.8 | 78.1 | 68.9 | 83.5 | 66.1 | 60.6 | 70.8 | 35.1 | 79.8 | 30.8 | 47.9 |
| | CSteer | 67.9 | 78.1 | 69.9 | 83.5 | 68.9 | 61.9 | 71.7 | 35.8 | 81.5 | 30.8 | 48.7 |
| Qwen3-VL 8B | *w/o* refer | 42.6 | 24.8 | 44.9 | 9.7 | 53.2 | 36.3 | 36.6 | - | - | - | - |
| | SoM | 70.9 | 76.4 | 70.2 | 77.4 | 69.3 | 64.4 | 71.5 | 41.0 | 88.7 | 40.0 | 55.9 |
| | CSteer | 70.7 | 75.8 | 73.1 | 80.6 | 76.1 | 71.9 | 74.7 | 43.3 | 89.5 | 41.5 | 57.5 |

GAR (Wang et al., 2025b) post-trained by GRPO with tailored contextual referring design. Such results suggest that recent general LMMs can recognize and reason with visual markers. Although our baseline being strong, we note that CSteer is effective in most cases, with improving both MC and OE tasks consistently.

**VIPBench** (Cai et al., 2024) covers multiple input domains and tasks, such as natural or optical images, with recognition, mathematical reasoning, and knowledge-enhanced questions. We present results of VIP-Bench in Tab. 2. Despite using natural images for vectoring (Peng et al., 2024), we still observe non-negligible gains over tasks like OCR or math reasoning (MAT), suggesting our design as an effective approach that could be generalized across input domains.

**BLINK** (Fu et al., 2024) is characterized by its comprehensive evaluation on visual cognition, including low-level cognition such as comparing reflectance from two referred points. We evaluated three subsets that involve visual referring (points), including relative reflectance, relative depth, and functional correspondence. The results are given in Tab. 2. We point out that even though we use recognition as the objective for vectoring where tasks are drastically different, we still observe gains on these tasks, indicating that our design is effective in tackling *multi-region visual referring*, mainly over comparing regions referred by points.

**INST-IT** (Peng et al., 2024) is a recent benchmark characterized by instance-level cognition, with both images and videos as testbeds. Both MC and OE questions are evaluated in our experiments. For videos, we set the input frames

*Table 3.* INST-IT Bench (Peng et al., 2024). The data is characterized by instance-level visual referring, including both multiple-choice and open-ended generation tasks. For evaluations of video subset, we use INST-IT-Video dataset to build vectors.

| Model | | Image | | Video | |
|---|---|---|---|---|---|
| | | OE | MC | OE | MC |
| Proprietary LMMs | | | | | |
| GPT-4o | | 74.1 | 84.8 | 65.5 | 81.0 |
| o3 | | 69.3 | 77.3 | 54.8 | 52.9 |
| Gemini-2.5-Pro | | 78.0 | 84.9 | 57.9 | 65.5 |
| Region LMMs | | | | | |
| VIP-LLaVA 7B | | 42.1 | 29.2 | - | - |
| Inst-IT-Qwen 7B | | 67.9 | 75.3 | 45.7 | 53.3 |
| General LMMs + Training-Free Methods | | | | | |
| InternVL-3.5 8B | *w/o* refer | 55.7 | 55.4 | 45.8 | 52.6 |
| | SoM | 73.7 | 79.6 | 50.3 | 60.6 |
| | CSteer | 74.2 | 80.0 | 51.9 | 61.8 |
| Qwen3-VL 8B | *w/o* refer | 24.7 | 61.8 | 43.9 | 41.4 |
| | SoM | 78.5 | 83.7 | 51.5 | 58.2 |
| | CSteer | 80.4 | 83.7 | 52.3 | 60.1 |

to 16 consistently and format our inputs with customized LMM prompts (details in the Appendix). For both image- and video-level referring that relate multiple regions, we observe consistent performance gains with our proposed approach, as presented in Tab. 3, indicating that our design could be generalized to capture spatio-temporal cues.

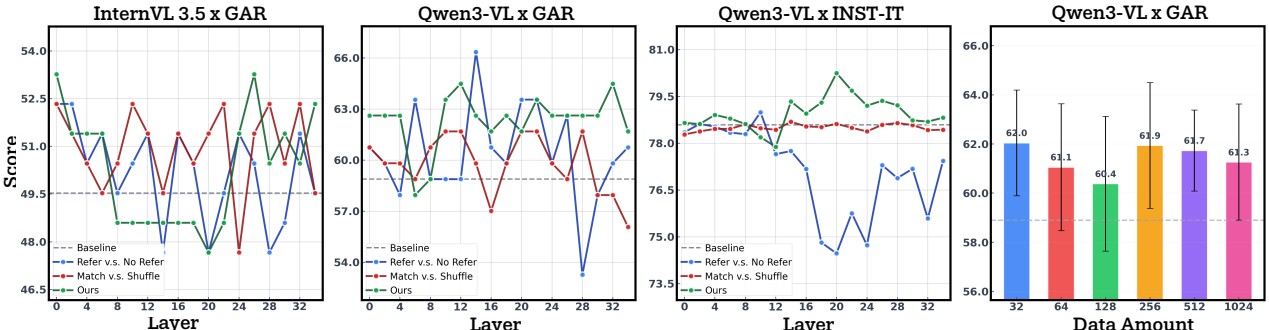

*Figure 5.* **Per Layer and Data Scale Results for Vectoring Approaches.** (*left*) We perform layer sweeps for both InternVL-3.5 (Wang et al., 2025c) and Qwen3-VL (Bai et al., 2025), reporting the performance on GAR (Wang et al., 2025b) and Inst-IT (Peng et al., 2024) (*right*) We apply different number of samples for vectoring, from 32 to 1024. Across data scales we observe consistent gains.

*Table 4.* **Ablation for Vectoring**.

| Design | InternVL 3.5 | | Qwen3-VL | |
|---|---|---|---|---|
| | GAR | INST | GAR | INST |
| baseline | 49.5 | 73.7 | 58.9 | 78.6 |
| *refer v.s. no refer* | 50.5 | 74.1 | 62.6 | 77.3 |
| *match v.s. shuffle* | 51.4 | 74.0 | 61.7 | 78.6 |
| *gt v.s. rollout* | 53.3 | 74.6 | 64.5 | 80.2 |
| *rewrite v.s. rollout* | 53.3 | 74.8 | 65.4 | 80.4 |

*Table 5.* **Ablation for Steering**.

| Design | InternVL 3.5 | | Qwen3-VL | |
|---|---|---|---|---|
| | GAR | INST | GAR | INST |
| baseline | 49.5 | 73.7 | 58.9 | 78.6 |
| *+ all* | 53.3 | 74.8 | 65.4 | 80.4 |
| *+ marker-only* | 53.3 | 74.0 | 63.6 | 79.8 |
| *+ in-query* | 55.1 | 74.2 | 64.5 | 80.4 |
| *+ ours* | 55.1 | 74.2 | 66.4 | 80.3 |

## 4.3. Ablation

We perform ablations from several aspects in detail to isolate key designs, including vectoring with referential rewrites and steering with decomposition, which ground on our observations that steering within multimodal query contexts behaves differently with that during decoding steps.

**Layer Sweep**. We commence with layer sweeps for both MCs and OEs with GAR-Bench (Wang et al., 2025b). For MC questions, we apply in-query steering, along with editing the last token, considering that general LMMs answer MCs with one token only. For OEs, we apply steering with both in-query and in-decoding. During decoding steps, we only apply steering to marker tokens. The layer sweep results for both MC and OE are given in Fig. 5. On one hand, we observe consistent gains across all layer settings, suggesting effectiveness of our design on contextual referring. On another, we observe different peakings for MCs and OEs, specifically, MCs favors early layers, while OEs favors middle or late layers instead.

**Manual Vectoring**. In Sec. 3, we introduce several designs regarding vector generation that involve contrasting signals, which can serve as editing directions to elicit contextual referring of general LMMs. For four designs that we have proposed, we find that the first two straightforward ones bring limited gains (*a.k.a.*, *refer v.s. no refer*, *exact natching v.s. marker shuffle*). We attribute this to the superiority of the baseline we use (*e.g.,* Qwen3-VL). For a strong baseline,

manual correction that guides contextual referring could be not effective, especially when LMMs inference with simple cases (*i.e.* all samplings are correct).

**Referential Rewrites**. As mentioned in Sec. 3, we only retain samples with false rollouts from our baselines for constructing vectors. Then, a judge LLM will correct false instance matching in localised captions (*i.e.*, rollouts) or add missing ones. We observe that such a design choice already has decent performance gains, as compared to previously mentioned approaches (given in Tab. 4). We then compare using different samples as positive for vectoring, specifically ground-truth captions and corrected rollouts re-written from generated false rollouts directly by a text-only LLM (Hurst et al., 2024). The results in Tab. 4 indicate that the rollout correction is slightly better for vectoring.

**Steering Options**. We include two design options that constitute CSteer here, *marker only* and *in-query steering*. We take (Rimsky et al., 2024) as our baseline, which applies the test-time intervention by Eq. 9 in decoding to all steps. The results are summarized in Tab. 5. We should point out that the vanilla approach with our contextual vectors has remarkable gains. Then, for marker only, instead of steering at all decoding steps, we only apply it when decoded tokens are markers, while other tokens remain unaffected. We observe occasional drops or no significant changes. After that, we apply steering to query tokens within contexts additionally and find that this often benefits.

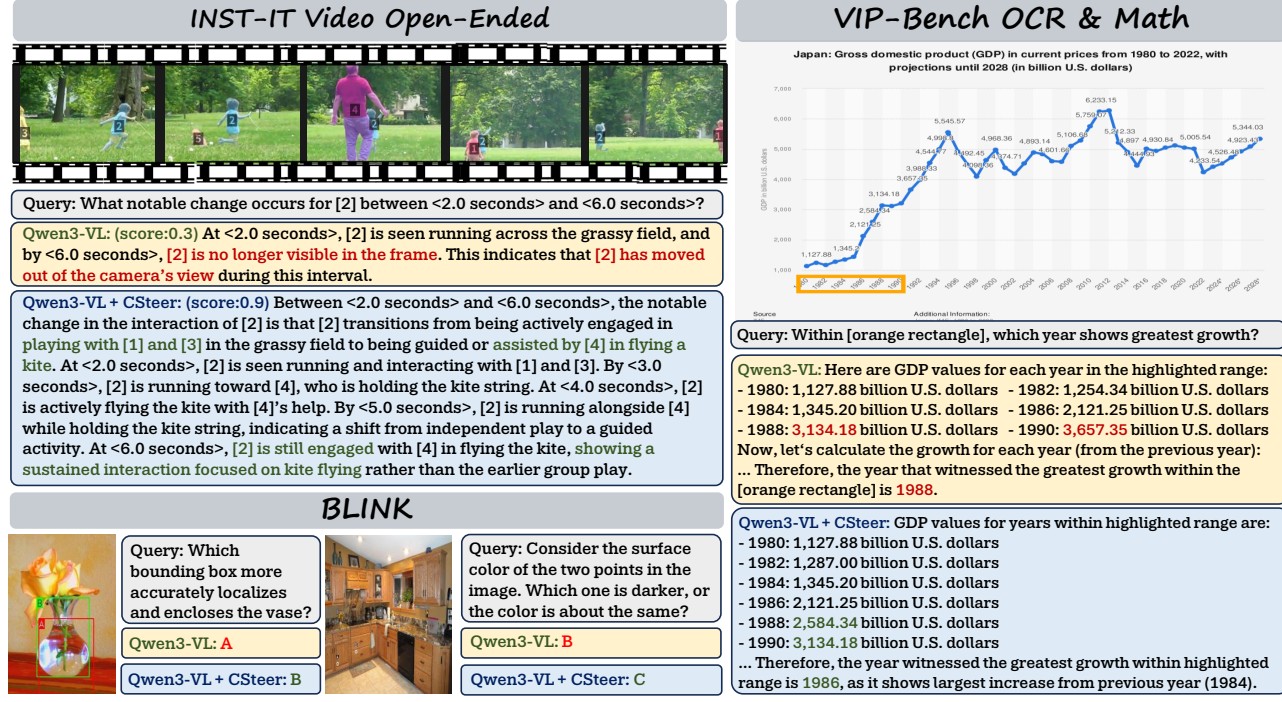

*Figure 6.* **Case Study**. We further show CSteer from INST-IT vectors addresses video localized captioning (Peng et al., 2024), reflectance comparison (Fu et al., 2024) and OCR reasoning (Cai et al., 2024).

**Decomposition**. Motivated by existing studies on LLM or LMM information flow (Wang et al., 2023; Jiang et al., 2025), which claim that LLM or LMM aggregates in early layers and predicts in late layers. This motivates us to separate these two options with different parameter settings, as illustrated in Fig. 4, where we get slightly better results.

### 4.4. Vectoring Data Scale

Considering vectoring, it is natural to consider how many data are sufficient for building reliable contextual referring vectors, and how many data will saturate. To answer this, we scale the data size from 32 to 1024, with different vectoring designs. Quantitative results with GAR as evaluation data are given in Fig. 5. It shows that gains for *rewrite w.r.t. rollout* are effective and robust across all data scale, even competitive with vectoring from fewer samples, which also indicating its efficiency.

### 4.5. Case Study

We present several cases where steering works in Fig. 6, including correcting the wrongly referred objects or regions or failing to recognize inter-object relations when objects referring are correct. Moreover, from a perspective of prompting format, we show that our approach is generalizable across points (*e.g.*, the relative reflectance example, where asking lighting condition on given points), boxes (*e.g.*, comparing which box better localizes and encloses a region, or rec-

ognize and reason over optical characters), and instance numerical identifiers. Lastly, we show video referring cases as well in Fig. 6 to show that our approach captures spatiotemporal cues as well on instance-level.

## 5. Conclusion

In this paper, we address the challenge of multi-region visual referring, with a novel training-free design named as **CSteer**. Instead of following typical approach that trains an individual encoder, we ground our analyzes from referring ambiguity in general LMMs when overlaying multiple markers on images, and more complex scenarios when referring questions are contextual. We perform careful ablations over vectoring and steering design choices, with sufficient layerwise sweep, where we find that visual referring could be elicited by contrasting false responses with their paired rewrites for vectoring, along with steering decomposition that involves in-query steering at early layers and decoding steering at middle or last.

**Limitations and Future Directions**. Due to our resource limitations, we cannot apply our design to larger scale general LMMs so far to test their efficacy, or test how CSteer may affect thinking with localization cues. We believe that our work will contribute to advancing the development of interactive LMMs in a training-free manner, or minimal adaptations with few data.

## Impact Statement

This work advances multi-region visual referring through a training-free approach that composes existing pre-trained models. By not requiring additional training data or fine-tuning, our method avoids introducing new biases or privacy concerns beyond those inherent in the foundation models used. The primary ethical considerations relate to the responsible use of the underlying pre-trained models, and we encourage practitioners to evaluate fairness and bias in their specific application contexts.

## Acknowledgement

This study is funded by the Ministry of Education, Singapore, under the Tier-2 project scheme with project number MOET2EP20123-0003.

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

# A. More Details

## A.1. Data for Building Vectors

We randomly sample our data for vectoring from INST-IT (Peng et al., 2024) training set. We point out that we use only two set of data across all our experiments, INST-IT-image and INST-IT-video. We use vectors from these two data sources to solve image and video scenarios, respectively.

## A.2. Rollout

For both Qwen3-VL-8B (Bai et al., 2025) and InternVL-3.5-8B (Wang et al., 2025c), we sample 8 responses with temperature set as 1.0. We then use a text-only LLM Qwen-2.5-72B (Team et al., 2024) to score all responses with ground-truth instance captions. For building *contextual vectors*, we abandon responses with scores higher than 0.6. We use the rest of rollouts for referential rewrites and generating contextual vectors.

# B. Relative Attention

Recent study (Zhang et al., 2025a) uses attentions from generic instruction (*e.g.*, write a general description about this image) to normalize attentions from given queries, which emphasizes semantically relevant information by attention maps. Mathematically, it is represented as,

$$A_{rel}(x, q) = \frac{A_{si}(x, q)}{A_{si}(x, q')} \tag{10}$$

In this way, LMMs know where to look and can perform automatic visual cropping (Zhang et al., 2025a). However, as observations in Fig. 2, relative attention is no longer effective for multi-region cases.

# C. More Related Works

We include more related papers here that better positioning our paper, which we elaborate as follows.

**Steering**. As a representative approach for improving contexts in LLM (Mei et al., 2025; Yu et al., 2026) or LMM (Li et al., 2025a; Xing et al., 2025; Yang et al., 2025) when test time, Contrastive Activation Addition (Rimsky et al., 2024) modifies LLM diverse behaviors such as refusal or hallucination mitigation through teacher forcing, and contrasting the last tokens. Through wide explorations they find that CAA is effective in guiding multiple LLM behaviors in training-free, such as refusal (Arditi et al., 2024; O'Brien et al., 2024), unlearning (Lynch et al., 2024; Shen et al., 2025), or hallucinations (Jesson et al., 2024), with recent papers extend this to multimodal scenarios (Liu et al., 2025; Li et al., 2025b; Su et al., 2025). The methodology of this approach has been greatly improved, including spectral editing (Qiu et al., 2024), angular control (Vu & Nguyen, 2025), or more fine-grained engineering with sparse autoencoders (O'Brien et al., 2024).

# D. Prompts

We list prompts we used for all our experiments for reimplementation purpose.

## D.1. Vectoring

### D.1.1. ROLLOUT

---

**INST-IT (IMAGE)**

<image>
Question: {question}
Prompt:
Task Definition:
You are an expert in image analysis. In this task, you will receive an image, and your task is to answer the given question based on the image content.
# Guidelines and Rules:
- Object References: In the image, each object has a unique ID. Use this ID in your response to specify objects, formatted as [ID] for a single object (e.g., "[8]") or as [ID1] [ID2] ... for multiple objects, such as "[3] [4] [5]". Avoid commas, ranges, or phrases like "[1, 2, 3]" or "[1] to [3]". The IDs in the images and questions match directly.

---

---

**INST-IT (VIDEO)**

<image>
Question: {question}
Prompt:
Task Definition:
You are an expert in video analysis. In this task, you will receive a series of frames as a video, and your task is to answer the given questions based on the video content.
# Input Format:
There are serveral images inputs as video frames. Each frame can be referenced by its timestamp (indicating when it appears in the video). For example, the first frame can be referred to as <1>.
# Guidelines and Rules:
- Object References: Each object has a unique ID. Use this ID in your response to specify objects, formatted as [ID] for a single object (e.g., "[8]") or as [ID1] [ID2] ... for multiple objects, such as "[3] [4] [5]". Avoid commas, ranges, or phrases like "[1, 2, 3]" or "[1] to [3]". The IDs in the images and questions match directly.
- Time References: Use timestamps to indicate moments or intervals in the video. For a specific moment, format as <timestamp> (e.g., "at <3>"). For an interval, use <start_timestamp>-<end_timestamp> (e.g., "during <5>-<7>"). Always enclose timestamps in <>.

---

## D.1.2. SCORING

**JUDGE(IMAGE)**

<image>
Prompt:
# Task Description:
You are an expert evaluator tasked with scoring the accuracy of responses to open-ended questions. You will be provided with a question, a ground-truth answer, and a response from a tester. Your job is to assess the accuracy of the response and provide ONLY a score between 0 and 1.

# Guidelines:
- Score Range: Your score must be between 0 and 1. A higher score means more correctness. Choose from: 0, 0.1, 0.2, 0.3, 0.4, 0.5, 0.6, 0.7, 0.8, 0.9, 1.0
- Consider the question, the ground-truth answer, and the tester's response together to determine correctness.
- Objects in questions and answers may be referenced using the format [ID] (e.g., [1], [2]). Ensure that any objects referenced in the tester's response match correctly with the ground-truth answer.

# Output Format:
IMPORTANT: You must output ONLY a number between 0 and 1 (e.g., 0.5, 0.7, 1.0). Do not output any text, explanation, or reasoning. Only output the numeric score.

# Input:
- Question: {question}
- Ground Truth Answer: {ground_truth}
- Response: {response}

---

**JUDGE(VIDEO)**

<image>
Prompt:
# Task Description:
You are an expert evaluator tasked with scoring the accuracy of responses to open-ended questions. You will be provided with a question, a ground-truth answer, and a response from a tester. Your job is to assess the accuracy of the response and provide ONLY a score between 0 and 1.

# Guidelines:
- Score Range: Your score must be between 0 and 1. A higher score means more correctness. Choose from: 0, 0.1, 0.2, 0.3, 0.4, 0.5, 0.6, 0.7, 0.8, 0.9, 1.0
- Consider the question, the ground-truth answer, and the tester's response together to determine correctness.
- Objects in questions and answers may be referenced using the format [ID] (e.g., [1], [2]). Ensure that any objects referenced in the tester's response match correctly with the ground-truth answer.
- Time points may be indicated with <timestamp> (e.g., <1>), and time intervals with <start_timestamp>-<end_timestamp> (e.g., <3>-<5>). Verify that the tester's response includes accurate time expressions.

# Output Format:
IMPORTANT: You must output ONLY a number between 0 and 1 (e.g., 0.5, 0.7, 1.0). Do not output any text, explanation, or reasoning. Only output the numeric score.

# Input:
question: {instruction}
ground truth answer for the question: {ground_truth}
response from the tester: {rollout_text}

---

## D.1.3. REWRITE

**REWRITER (Image)**

Prompt:
# Task Description:
You are an expert text editor. Your task is to rewrite an incorrect response to make it more accurate and aligned with the ground truth answer, while maintaining the original response's style and structure as much as possible.

# Context:
- Question: {question}
- Ground Truth Answer: {ground_truth}
- Original Response (to be rewritten): {rollout_text}

# Guidelines:
1. **Accuracy First**: The rewritten response must be factually correct and aligned with the ground truth answer.
2. **Object References**: Objects in the question and ground truth may be referenced using the format [ID] (e.g., [1], [2]). Ensure that any objects referenced in the rewritten response match correctly with the ground truth answer.
3. **Preserve Structure**: Try to maintain the original response's structure, style, and length as much as possible, while correcting errors.
4. **Semantic Alignment**: The rewritten response should convey the same meaning as the ground truth, but you can use different phrasings or synonyms as long as the meaning is preserved.
5. **Complete Response**: Ensure the rewritten response is complete and addresses the question fully.

# Output Format:
IMPORTANT: Output ONLY the rewritten response text. Do not include any explanations, reasoning, or meta-commentary. Just output the corrected response that should score 0.7 or higher when evaluated against the ground truth.

# Rewritten Response:

---

---

**REWRITER (Video)**

Prompt:
# Task Description:
You are an expert text editor. Your task is to rewrite an incorrect response to make it more accurate and aligned with the ground truth answer, while maintaining the original response's style and structure as much as possible.

# Context:
- Question: {question}
- Ground Truth Answer: {ground_truth}
- Original Response (to be rewritten): {rollout_text}

# Guidelines:
1. **Accuracy First**: The rewritten response must be factually correct and aligned with the ground truth answer.
2. **Object References**: Objects in the question and ground truth may be referenced using the format [ID] (e.g., [1], [2]). Ensure that any objects referenced in the rewritten response match correctly with the ground truth answer.
3. **Time References**: Time points may be indicated with <timestamp> (e.g., <1>), and time intervals with <start_timestamp>-<end_timestamp> (e.g., <3>-<5>). Verify that the rewritten response includes accurate time expressions if needed.
4. **Preserve Structure**: Try to maintain the original response's structure, style, and length as much as possible, while correcting errors.
5. **Semantic Alignment**: The rewritten response should convey the same meaning as the ground truth, but you can use different phrasings or synonyms as long as the meaning is preserved.
6. **Complete Response**: Ensure the rewritten response is complete and addresses the question fully.

# Output Format:
IMPORTANT: Output ONLY the rewritten response text. Do not include any explanations, reasoning, or meta-commentary. Just output the corrected response that should score 0.6 or higher when evaluated against the ground truth.

# Rewritten Response:

## D.2. Evaluation

D.2.1. INFERENCE

---

**GAR (Multiple Choice)**

<image>
Question: {question}
Options:
{options}
Prompt:
Based on the image, select the best answer to the following multiple-choice question. Each relevant object is highlighted with a bounding box and a numeric ID, and objects are referenced in the question and options using the format [ID] (e.g., "[0]", "[1]"). Respond with only the letter (A, B, C, or D) of the correct option.

The best answer is:

---

## GAR (Simple Open-Ended)

<image>
Question: {question}
Prompt:
# Task Definition:
You are an expert in image analysis. In this task, you will receive an image, and your task is to answer the given question based on the image content.
# Guidelines and Rules:
- Object References: In the image, each object is surrounded by a box and has an unique ID. Use this ID in your response to specify objects, formatted as [ID] for a single object (e.g., "[8]") or as [ID1] [ID2] ... for multiple objects, such as "[3] [4] [5]". Avoid commas, ranges, or phrases like "[1, 2, 3]" or "[1] to [3]". The IDs in the images and questions match directly.
# Relation:
When describing spatial relations among objects, please consider multiple perspectives, including left-or-right, front-or-back, and other potential relations.
# Output Instructions:
Please first briefly recognize the referred objects or regions, then answer.
# Examples:
[0] is a person with a red hat who sits next to [1], a bird.
[1] is a cow standing on grass, in front of [0], a person taking photos for [1].

Based on the input image, please answer the question:

## GAR (Detailed Open-Ended)

<image>
Question: {question}
Prompt:
# Task Definition:
You are an expert in image analysis. In this task, you will receive an image from, where each relevant object is highlighted with a bounding box and a numeric ID overlaid on the image.
# Guidelines and Rules:
- Object References: In the image, each object has a unique ID that is displayed next to its bounding box. Use this ID in your response to specify objects, formatted as [ID] for a single object (e.g., "[0]") or as [ID1] [ID2] ... for multiple objects, such as "[0] [1] [2]". Avoid commas, ranges, or phrases like "[0, 1, 2]" or "[0] to [2]". The IDs in the images and questions match directly.

Based on the input image, please answer the question:

## VIP-Bench (Image Open-Ended)

<image>
Question: {question}
Prompt:
# Task Definition:
You are an expert in image analysis. In this task, you will receive an image, and your task is to answer the given question based on the image content.
# Guidelines and Rules:
- Object References: In the image, each object has a unique ID. Use this ID in your response to specify objects, formatted as [ID] for a single object (e.g., '[red box]', '[yellow box]'). The IDs in the images and questions match directly.

Based on the input image, please answer the question:

## BLINK (Multiple Choice)

<image>
Question: {question}
Options:
{options}
Prompt:
Based on the image, select the best answer to the following multiple-choice question. In both the question and options, specific objects are represented using the format [ID] (e.g., '[REF]', '[A]'). Respond with only the letter (A, B, C, or D) of the correct option.

The best answer is:

---

**INST-IT (Image Open-Ended)**

<image>
Question: {question}
Prompt:
# Task Definition:
You are an expert in image analysis. In this task, you will receive an image, and your task is to answer the given question based on the image content.
# Guidelines and Rules:
- Object References: In the image, each object has a unique ID. Use this ID in your response to specify objects, formatted as [ID] for a single object (e.g., "[8]") or as [ID1] [ID2] ... for multiple objects, such as "[3] [4] [5]". Avoid commas, ranges, or phrases like "[1, 2, 3]" or "[1] to [3]". The IDs in the images and questions match directly.

Based on the input image, please answer the question:

---

**INST-IT (Image Multiple Choice)**

<image>
Question: {question}
Options:
{options}
Prompt:
Based on the image, select the best answer to the following multiple-choice question. In both the question and options, specific objects are represented using the format [ID] (e.g., '[1]', '[2]'). Respond with only the letter (A, B, C, or D) of the correct option.

The best answer is:

---

**INST-IT (Video Open-Ended)**

<image>
Question: {question}
Prompt:
# Task Definition:
You are an expert in video analysis. In this task, you will receive a series of frames as a video, and your task is to answer the given questions based on the video content.
# Input Format:
There are serveral images inputs as video frames. Each frame can be referenced by its timestamp (indicating when it appears in the video). For example, the first frame can be referred to as <1>.
# Guidelines and Rules:
- Object References: Each object has a unique ID. Use this ID in your response to specify objects, formatted as [ID] for a single object (e.g., "[8]") or as [ID1] [ID2] ... for multiple objects, such as "[3] [4] [5]". Avoid commas, ranges, or phrases like "[1, 2, 3]" or "[1] to [3]". The IDs in the images and questions match directly.
- Time References: Use timestamps to indicate moments or intervals in the video. For a specific moment, format as <timestamp> (e.g., "at <3>"). For an interval, use <start_timestamp>-<end_timestamp> (e.g., "during <5>-<7>"). Always enclose timestamps in <>.

Based on the input video, please answer the question:

---

**INST-IT (Video Multiple Choice)**

<image>
Question: {question}
Options:
{options}
Prompt:
Based on the video, select the best answer to the following multiple-choice question. In both the question and options, specific objects are represented using the format [ID] (e.g., "[1]", "[2]"), and time references are shown using the format <timestamp> (e.g., "at <6>" or "during <7>-<8>"). Respond with only the letter (A, B, C, or D) of the correct option.

The best answer is:

D.2.2. JUDGE

---

**GAR (Simple Open-Ended)**

<image>
Prompt:
You are a language model expert. Your task is to evaluate the correctness of the model's output based on the provided ground truth and given masks.

- Ground truth: "{answer}"
- Model Output: "{model_output}"

Please determine if the model's output conveys the same meaning as the provided ground truth. If the output is semantically correct, return "True", otherwise return "False".

Attention:
1. The ground truth and model output do not need to match exactly, as long as they convey the same meaning. Synonyms and different phrasings are acceptable.

2. Do not output any reasoning. Do not perform correction. Please output only "True" or "False".

---

**GAR (Detailed Open-Ended)**

<image>
Prompt:
You are a language model expert. Your task is to evaluate the following model output based on the provided images, and subject, object, and relationship.

- subject_name: {subject_name}
- object_name: {object_name}
- predicate_name: {predicate_name}
- model_output: {model_output}

Task:
1. Check if the model output describes the {subject_name}.
2. Check if the model output conveys the relationship between {subject_name} and {object_name} related to {predicate_name}.

Note:
- The first task only requires checking if {subject_name} is mentioned in the model output.
- The second task asks if the output conveys a relationship related to {predicate_name} between {subject_name} and {object_name}, even if different words or phrases are used.
- If both tasks are successfully completed, return "True" Otherwise, return "False"
- Do not output any reasoning. Do not perform correction. Please output only just one "True" or "False".

**VIP-Bench (Image Open-Ended)**

<image>
Prompt:
Compare the ground truth and prediction from AI models, to give a correctness score for the prediction. <AND> in the ground truth means it is totally right only when all elements in the ground truth are present in the prediction, and <OR> means it is totally right when any one element in the ground truth is present in the prediction. The correctness score is 0.0 (totally wrong), 0.1, 0.2, 0.3, 0.4, 0.5, 0.6, 0.7, 0.8, 0.9, or 1.0 (totally right). Just complete the last space of the correctness score.

Question | Ground truth | Prediction | Correctness
— | — | — | —
What is x in the equation within the yellow rectangle? | -1 <AND> -5 | x = 3 | 0.0
What is x in the equation within the yellow rectangle? | -1 <AND> -5 | x = -1 | 0.5
What is x in the equation within the yellow rectangle? | -1 <AND> -5 | x = -5 | 0.5
What is x in the equation within the red rectangle? | -1 <AND> -5 | x = -5 or 5 | 0.5
What is x in the equation within the orange rectangle? | -1 <AND> -5 | x = -1 or x = -5 | 1.0
Can you explain this meme within the blue rectangle? | This meme is poking fun at the fact that the names of the countries Iceland and Greenland are misleading. Despite its name, Iceland is known for its beautiful green landscapes, while Greenland is mostly covered in ice and snow. The meme is saying that the person has trust issues because the names of these countries do not accurately represent their landscapes. | The meme talks about Iceland and Greenland. It's pointing out that despite their names, Iceland is not very icy and Greenland isn't very green. | 0.4
Can you explain this meme within the blue rectangle? | This meme is poking fun at the fact that the names of the countries Iceland and Greenland are misleading. Despite its name, Iceland is known for its beautiful green landscapes, while Greenland is mostly covered in ice and snow. The meme is saying that the person has trust issues because the names of these countries do not accurately represent their landscapes. | The meme is using humor to point out the misleading nature of Iceland's and Greenland's names. Iceland, despite its name, has lush green landscapes while Greenland is mostly covered in ice and snow. The text 'This is why I have trust issues' is a playful way to suggest that these contradictions can lead to distrust or confusion. The humor in this meme is derived from the unexpected contrast between the names of the countries and their actual physical characteristics. | 1.0

IMPORTANT: Output ONLY a single number between 0.0 and 1.0. No other text.

**INST-IT (Image Open-Ended)**

<image>
Prompt:
# Task Description:
You are an expert evaluator tasked with scoring the accuracy of responses to open-ended questions. You will be provided with a set of questions, each with a corresponding ground-truth answer, as well as responses from a tester. Your job is to assess the accuracy of each response and provide a score between 0 and 1.
# Guidelines:
- Score Range: Your score for each test item must be between 0 and 1. A higher score means more correctness. Choose from the following: 0 (completely incorrect), 0.1, 0.2, 0.3, 0.4, 0.5, 0.6, 0.7, 0.8, 0.9, 1.0 (completely correct)
- For each test item, consider the question, the ground-truth answer, and the tester's response together to determine correctness.
- Objects in questions and answers may be referenced using the format [ID] (e.g., [1], [2]). Ensure that any objects referenced in the tester's response match correctly with the ground-truth answer.
# Input Format:
The input is a set of test items to be scored, where each item includes:
- 'question';
- 'ground truth answer for the question';
- 'response from the tester'.
Now, let's begin the evaluation, here are the input test items:

---

**INST-IT (Video Open-Ended)**

<image>
Prompt:
# Task Description:
You are an expert evaluator tasked with scoring the accuracy of responses to open-ended questions. You will be provided with a set of questions, each with a corresponding ground-truth answer, as well as responses from a tester. Your job is to assess the accuracy of each response and provide a score between 0 and 1.
# Guidelines:
- Score Range: Your score for each test item must be between 0 and 1. A higher score means more correctness. Choose from the following: 0 (completely incorrect), 0.1, 0.2, 0.3, 0.4, 0.5, 0.6, 0.7, 0.8, 0.9, 1.0 (completely correct)
- For each test item, consider the question, the ground-truth answer, and the tester's response together to determine correctness.
- Objects in questions and answers may be referenced using the format [ID] (e.g., [1], [2]). Ensure that any objects referenced in the tester's response match correctly with the ground-truth answer.
- Time points may be indicated with <timestamp> (e.g., <1>), and time intervals with <start_timestamp>-<end_timestamp> (e.g., <3>-<5>). Verify that the tester's response includes accurate time expressions.
# Input Format:
The input is a set of test items to be scored, where each item includes:
- 'question';
- 'ground truth answer for the question';
- 'response from the tester'.
Now, let's begin the evaluation, here are the input test items:

## E. Data

We list the details of our evaluation data for reimplementation purpose.

*Table 6.* **Specifics for Evaluation Data.**

| Data | Input Type | QA Type | Subset | Full Name | Amount |
|------|-----------|---------|--------|-----------|--------|
| GAR-Bench (Wang et al., 2025b) | Image | MC | CLR | Color | 69 |
| | | | SHP | Shape | 64 |
| | | | TXT | Texture | 29 |
| | | | MAT | Material | 36 |
| | | | POS | Position | 64 |
| | | | NET | Non-Entity | 61 |
| | | | REL | Relation | 101 |
| | | Open-Ended | SIM | Simple | 97 |
| | | | DET | Detailed | 107 |
| VIP-Bench (Cai et al., 2024) | Image | Open-Ended | REC | Recognition | 253 |
| | | | OCR | Optical Character | 90 |
| | | | KNO | Knowledge | 60 |
| | | | MAT | Math | 31 |
| | | | REL | Relation | 41 |
| | | | LAN | Language | 22 |
| INST-IT (Peng et al., 2024) | Image | MC | | | 1,036 |
| | | Open-Ended | | | 1,036 |
| | Video | MC | | | 1,001 |
| | | Open-Ended | | | 1,001 |
| BLINK (Fu et al., 2024) | Image | MC | RRF | Relative Reflectance | 268 |
| | | | RDP | Relative Depth | 248 |
| | | | FCO | Functional Correspondence | 260 |

