# OpenReview forum: "Referring Multiple Regions with Large Multimodal Models via Contextual Latent Steering"
_ICML.cc/2026/Conference — ICML 2026 regular_

### Official Review · Reviewer_ZVwU · 2026-03-11

**Soundness:** 2
**Presentation:** 2
**Significance:** 2
**Originality:** 2
**Overall Recommendation:** 2
**Confidence:** 3

**Summary:**

This paper addresses the shortcomings of current LMMs in understanding multiple regions and proposes a training-free method, Contextual Latent Steering. By constructing positive–negative sample pairs, CSteer computes a contextual vector to enhance the model’s attention to the referring region, and achieves certain performance improvements on both InternVL3.5 and Qwen3-VL. The paper also compares different methods for constructing contrastive sample pairs and analyzes the impact of different steering options on performance.

**Compliance With Llm Reviewing Policy:**

Affirmed.

**Final Justification:**

Thank you for the rebuttal and clarifications. While I appreciate the effort, my concerns regarding clarity, methodological grounding, and novelty remain.

Key definitions are still unclear, important details affecting reproducibility are under-specified, and the link between the method design and the claimed multi-region capability is not well established. The conceptual distinction from prior work also remains limited.

Overall, these issues would require substantial revisions beyond the camera-ready stage. Therefore, I maintain my original score.

**Key Questions For Authors:**

See Weakness

**Limitations:**

yes

**Strengths And Weaknesses:**

**Strengths:**
1. The problem of insufficient multiple-region understanding is meaningful.
2. The proposed method is training-free.
3. The steering vector perspective makes sense.

**Weaknesses:**
1. The construction of the contextual vector is not described concretely enough and is too conceptual, making it difficult to map to the specific operations inside the LMM. For example, the specific meanings of $f_{+}$ and $f_{-}$ in the LMM context are unclear, it is not clear which internal representation $h$ corresponds to in the model, and $x_{+}$ and $x_{-}$ lack concrete definitions under the LMM setting.
2. The paper does not describe how the data used to construct the contextual vector are obtained, and how they are aggregated into the final vector.
3. On Inst-Bench, the improvement over SoM is too small.
4. The paper claims to address the multi-region understanding problem in LLM visual referring, but it does not explain the connection between CSteer and multi-region understanding. From the description in the Method section, I do not perceive any design specifically targeting the multi-region setting.
5. Some figures are placed too far from the corresponding text, such as Figure 4. Figure 3 does not have a caption, and the right half is difficult to understand. The pink contextual vector does not appear to interact with any other component.
6. Is the input sequence received by CSteer Eq. (2) or Eq. (3)? According to the description in lines 214–216, it should be Equation (3), but this is not explicitly stated in the paper.
7. The novelty is limited. CSteer directly applies CAA[1] to LMMs and modifies the computation of the steering vector for the image modality, but there is no fundamental difference.

Reference
[1] ”Steering Llama 2 via Contrastive Activation Addition” in ACL 2024

---

> ### Author Rebuttal · Authors · 2026-03-31
>
> We thank the reviewer for the detailed feedback. Please find our clarifications and responses to mentioned concerns as follows.
> **W1 Vector**. **(a) Symbol**. We kindly point out that x_+ and x_- are both multimodal inputs, while f_+ and f_- mean two forward passes for x_+ and x_-, respectively. We could make it clearer in the updated manuscript. **(b) Contextual Vector**. As mentioned in Line 203 (left), the last activations of inputs for given layers are derived for computing editing directions. **(c) Representation h**. With respect, we note that we do not use h, but use h_t^l across our manuscript. With token position t and layer l, each token could be located accurately. Ignoring it would make it unclear.
> **W2 Data**. **(a) Vector Data**. We refer the reviewer to Appendix A.1, where we describe the data source we use to build the contextual vectors. **(b) Aggregation**. The final vector is obtained by simply averaging across vectors from all data.
> **W3 INST-IT**. While the gains on this benchmark are relatively moderate compared to other tests (i.e., with Qwen3-VL, 1.1% on INST-IT v.s. 3.4% on GARBench), we note that the baseline is strong (we use Qwen3-VL and InternVL-3.5) and GARBench is a more recent benchmark (public in 2025.10, while INST-IT is public in 2024.12). On GARBench, we can achieve a 7.5% gain on detailed captioning (58.9 to 66.4). Considering multiple evaluations, we believe the gains are non-trivial.
> **W4 Motivation**. As mentioned in Line 093-101 (left), CSteer addresses multi-region understanding by correcting erroneous localized captions from general LMMs, such as rectifying mismatched numerical identifiers or complementing missing regions. Unlike explicit multi-region encoding designs, CSteer corrects general LMMs false multi-region behaviors. The motivation is recognized by all other reviewers.
> **W5 Presentation**. We thank the reviewer for pointing out the presentation issues, which could improve the clarity. **(a) Figure 3**. We will add the missing caption for Figure 3 as given in https://anonymous.4open.science/r/csteer-2119/Reb2_Fig3_vector.png, improve the clarity of the visualization. **(b) Pink contextual vectors**. For describing, the pink contextual vectors in Figure 3 intervenes LMM inference, as illustrated in Figure 4.
> **W6 Input Sequence**. We mentioned in Line 172-173 that Eq. (2) is for tailored training approaches. We will state more clearly in our paper that we follow Eq. (3).
> **W7 Comparison to CAA**. We respectfully point out that we never claimed CAA, the contrastive approach, is one of our contributions. We would like to justify our novelty in two dimensions,
> a) Considering closely related literature (i.e., visual object referring, contextual referring, multi-region referring), this is the first work that follows steering to address visual referring tasks.
> b) In some cases, naive designs derived from CAA are not effective for referring. For example, in Figure 5 (Qwen3-VL on INST-IT), no performance gains can be observed when applying two naive strategies for constructing vectors (Refer v.s. No Refer, Match v.s. Shuffle).

---

> > ### Author Rebuttal · Reviewer_ZVwU · 2026-04-04
> >
> > Thank you for your rebuttal. I appreciate the effort to clarify several points. Below are my follow-up comments.
> >
> > * Definition. Thank you for clarifying that ($x_+$) and ($x_-$)are multimodal inputs, and that ($f_+$) and ($f_-$) denote forward passes. However, my concern remains that these elements are not clearly defined in the paper itself. Key notations such as ($h_t^l$) should be explicitly introduced and grounded, rather than left for the reader to infer.
> >
> > * Data. You refer to Appendix A.1 for details. However, for a training-free method, the data used to construct the vectors directly affects the results. Important factors such as sample size, variability across samples, and potential randomness are not sufficiently specified, which raises concerns about reproducibility and stability.
> >
> > * Motivation for multi-region. The rebuttal explains that CSteer addresses multi-region understanding by correcting erroneous localized captions. However, I still do not see a design element that explicitly targets multi-region reasoning. A generic steering mechanism applied to single-region referring could plausibly yield similar behavior. Are single-region results available? The connection between the method and the claimed multi-region capability remains unclear.
> >
> > * Comparison to CAA. My point was not whether the paper claims CAA as a contribution, but that the proposed method appears to be a relatively direct adaptation of prior CAA approaches to LMMs, with limited conceptual distinction. This concern has not been substantively addressed.
> >
> > Overall, while I appreciate the clarifications provided in the rebuttal, my main concerns regarding clarity, methodological grounding, and novelty remain. Therefore, my overall recommendation is unchanged.

---

> > > ### Author Response · Authors · 2026-04-04
> > >
> > > Thank you for responses from the reviewer. Please see the follow-up clarifications as follows,
> > >
> > > **(a) Definition**. As the reviewer suggests, we believe these should be addressed in our updated version. This could improve the clarity of this paper.
> > >
> > > **(b) Data**. We reply to your follow-up concerns point-by-point as follows,
> > >
> > > **(b-1) Sample Size**. Please refer to Fig. 5 in our manuscript, where we presented these results, showing that gains are consistent regardless of sample size (across 32/64/128/256/512/1024).
> > >
> > > **(b-2) Sample Source**. Please refer to our rebuttal replies to the reviewer **sKs7 Q1**, where we used **pixelrefer-mdvp** and **pixelrefer-osprey** as source and provided the results.
> > >
> > > **(b-3) Potential Randomness**. We kindly point out that we do not understand this very clearly. If the randomness means **random seed** (if understand correctly), we use greedy decoding for all our test data, there is no randomness regarding sampling temperature.
> > >
> > > **(c/d) Motivation and Comparison**. For the mentioned multi-region design, we believe the reviewer means explicit modules for multiple regions. We kindly point out that a large portion of **one-region** problems are in fact **multi-region**, a.k.a. **contextual**. Like the case in lower-right (the third one) in Figure 2, there is only one region mentioned in question, but it is actually a multi-region problem (needs to fully understand the image, to know [3] and [4] in the same lane, and [2] is not). This is why our method focuses on **implicit** behavior by **steering**, rather than explicit design for multiple regions.
> > >
> > > Overall, we think it is good to have disagreements over our contributions as they makes our arguments more clear. We thank the reviewer for the constructive suggestions. Hope our follow-up replies have addressed some of the concerns.

---

### Official Review · Reviewer_mxzG · 2026-03-12

**Soundness:** 3
**Presentation:** 2
**Significance:** 2
**Originality:** 2
**Overall Recommendation:** 3
**Confidence:** 4

**Summary:**

This paper proposes a new training-free latent steering approach to improve VLM's performance on multiple-region referring tasks. The authors construct steering vectors from paired (correct/incorrect) responses obtained via different approaches. The authors show that this approach outperforms baselines across datasets that require multi-region referring capabilities, demonstrating the efficacy of the proposed approach. Moreover, the authors perform ablations to show the effectiveness of their design choices.

**Compliance With Llm Reviewing Policy:**

Affirmed.

**Final Justification:**

I thank the authors for their rebuttal. I would specifically like to appreciate their effort in spending credits on industrial APIs to improve the quality of their work with additional results. Concerns W2-5 from my original review have been appropriately addressed.

However, one of my concerns regarding the steering vector design choice remains. The proposed method does not clearly specify how to handle different types of errors arising from the two scenarios, such as marker shuffle or no refer. Additionally, the manuscript, in my opinion, needs significant improvement, as mentioned in my original review and noted by another reviewer (ZVwU).

Due to the above reasons, I believe the paper cannot be accepted in its current state. I wish the authors good luck with their submission!

**Key Questions For Authors:**

**Please see the concerns and questions raised in the Weaknesses above**

Additionally,

**Q1.** What does *"layer sweeps"* in L363 mean? Does this mean that the authors perform steering on that particular layer?

**Q2.** In L218R the authors mention that **Marker-only** strategy leads to *"minimal impact"*, but why does that happen?

**Limitations:**

yes

**Strengths And Weaknesses:**

### Strengths

**S1.** *Training-free approach*: In contrast to most baselines that perform region-referring, the authors solve this problem using training-free latent steering, which leads to significantly lower compute budget.

**S2.** *Recent baselines*: The authors report the efficacy of the proposed approach on two recent baselines -- Qwen 3 VL and Intern-3.5 VL -- and show improvements over both, demonstrating that their approach is effective even for recent VLMs.

**S3.** *Error analysis and Motivation*: I appreciate the author's effort in performing error analysis of the current VLMs in **Figure-2** to motivate their work. In general, I feel the work's use case is well motivated.

### Weaknesses

**W1.** *Averaging of latent directions*: The authors propose to construct the steering direction (vector) using different approaches mentioned in section 3.2. However, in my opinion, using an average direction (assuming that's what the authors do as there is no clear description as to how the vectors obtained from different methods are combined) makes the impact of a single behavior less. For example, consider creating a vector using the ***Refer vs No Refer (RvNR)*** method. Steering the outputs in this direction will force the latents to produce the outputs which have referring (instead of having no referring). However, if we construct the direction using ***Exact matching vs Marker shuffle (EMvMS)***, this will make the outputs have correct marker positions instead of having shuffled markers. These two vectors obtained from RvNR and EMvMS methods are in two different directions. So, taking an average of these does not guarantee the necessary multi-region referring behavior.


**W2.** *Implementation details are missing*: The authors fail to provide the details on what data they use to construct the steering vectors. If they use a small portion of the test data to construct these, it is a severe bottleneck. Moreover, as mentioned above, the authors do not provide details on how they aggregate the vectors obtained from the different methods described in section 3.2. Moreover, the authors do not specify the metrics they use to present their results.

**W3.** *Baseline model results missing*: In tables 1, 2, and 3, the results for base InternVL 3.5 and Qwen3-VL are missing. The authors provide only results after using SoM; what are the results when simply passing the query into the dataset to these models?

**W4.** *Comparison to more inference time methods*: The authors can also compare to some other inference-time methods for faithful VLM outputs, such as VCD [1], ICD [2], etc. While these methods do not directly tackle multi-region referring, they are inference-time approaches, and it would be interesting to see if they help in any way to tackle the task of visual referring.

**W5.** **Different baselines for different tables**: Tables 1, 2, and 3 have different baselines reported. I understand that some baselines might not have code available, but at least for the proprietary LMMs, I would expect to see the same baselines across all three tables; the results are missing. Additionally, the authors report Gemini-2.5-Pro results in Table 1 but use Gemini 1.5 in Table 3. This makes the study incomplete and inconsistent.

**W6.** *Paper writing issues*: I would not consider this a major weakness, but the paper still needs multiple changes to be camera-ready.
- Spelling mistakes: *"Genimi-2.5-Pro"* (L270L), *"natching"* (L382L) (there are more -- please check).
- Figure/Table captions: The captions to figures and tables are very incomplete, and it is hard to infer the purpose of the table/figure from just the caption. The captions are supposed to be self-sufficient for academic publications. (See Fig. 3, 5 and Tab. 1, 3, 4, 5).
- Figure-3 is hard to follow. What is the purpose of the figure? I would recommend restructuring or providing details in the caption.
- The authors do not explicitly mention what *SoM* in Tables 1-3 means.

[1] Leng, Sicong, Hang Zhang, Guanzheng Chen, Xin Li, Shijian Lu, Chunyan Miao, and Lidong Bing. "Mitigating object hallucinations in large vision-language models through visual contrastive decoding." In Proceedings of the IEEE/CVF Conference on Computer Vision and Pattern Recognition, pp. 13872-13882. 2024.

[2] Wang, Xintong, Jingheng Pan, Liang Ding, and Chris Biemann. "Mitigating hallucinations in large vision-language models with instruction contrastive decoding." In Findings of the Association for Computational Linguistics: ACL 2024, pp. 15840-15853. 2024.

---

> ### Author Rebuttal · Authors · 2026-03-31
>
> We appreciate recognition from the reviewer, regarding training-free approach, recent baselines, and motivation. Please see our point-by-point replies.
> **W1 Clarification on Averaging**. We kindly point out a potential misunderstanding that the four approaches mentioned in section 3.2 (Refer vs No Refer, Exact Matching vs Marker Shuffle, GT vs Rollout, Rewrite vs Rollout) are alternative, mutually exclusive designs, rather than components that are combined together. In practice, each approach independently defines a contrastive setting, and builds its own vectors. We do not average vectors across designs.
> **W2 Data Details**. **(a) Data for Steering Vectors**. We refer the reviewer to Appendix A.1, where we describe our implementations for this. We should note that data for constructing vectors is not from test data, but from INST-IT train data.
> **(b) Details on Evaluation Metrics**. We will include this in the updated manuscript for clarity. Basically, we follow the official metrics from the benchmarks we use, using accuracy for multiple choice questions and GPT-assisted scoring for open-ended ones. Judges prompts for each evaluation are given in Appendix D.
> **W3 Baseline**. The results are as follows. We point out that the query without set-of-mark prompting becomes ambiguous (examples in https://anonymous.4open.science/r/csteer-2119/Reb4_case.png), where LMMs are expected to refuse to answer, and we find Qwen3-VL are more frequent to refuse.
> | Benchmark | Metric | InternVL-3.5 (new) | Qwen3-VL (new) |
> |---|---|---:|---:|
> | GAR | CLR | 24.6 | 31.9 |
> | GAR | SHP | 50.0 | 43.7 |
> | GAR | TEX | 48.3 | 34.5 |
> | GAR | MAT | 30.6 | 44.4 |
> | GAR | POS | 43.8 | 45.3 |
> | GAR | NET | 24.6 | 26.2 |
> | GAR | REL | 44.6 | 45.5 |
> | GAR | ALL | 38.2 | 39.3 |
> | GAR | SIM | 27.8 | 24.7 |
> | GAR | DET | 46.7 | 39.2 |
> | VIP | AVG | 49.6 | 36.6 |
> | INST-IT IMAGE | MC | 55.4 | 61.8 |
> | INST-IT IMAGE | OE | 55.7 | 24.7 |
> | INST-IT VIDEO | MC | 52.6 | 41.4 |
> | INST-IT VIDEO | OE | 45.8 | 43.9 |
> ****
> **W4 VCD/ICD**. We agree these inference-time, hallucination mitigation approaches potentially benefit multi-region visual referring. To compare against them, we test mentioned approaches on GAR-Bench (DET). Results are summarized in a table here. Note that we use greedy decoding across our paper while cd approaches normally test on sampling setup (temperature not 0). We give both results here and will update this in our manuscript.
> | | Qwen3-VL | w. VCD (new) | w. ICD (new) | w. CSteer |
> |---|---:|---:|---:|---:|
> | Greedy | 58.9 | 57.0 | 55.1 | 66.4 |
> | Sample | 54.2 | 57.0 | 59.8 | 62.6 |
>
> **W5 Proprietary**. We agree with the suggestion that completes our results. Please find the results here, where we include consistent results across Table 1,2,3 for three models (Gemini-2.5-Pro, O3 and InternVL3-78B). For Gemini 1.5 results in Table 3 from INST-IT paper [Peng et al., 2024], we will update them with Gemini-2.5-Pro ones.
> | Benchmark | Metric | Gemini-2.5-Pro | O3 | InternVL3-78B |
> |---|---|---:|---:|---:|
> | VIP | AVG | 82.9 | 73.1 | 75.0 |
> | BLINK | RRF | 61.2 | 51.5 | 29.9 |
> | BLINK | RDP | 83.1 | 76.6 | 83.1 |
> | BLINK | FCO | 69.2 | 65.3 | 45.4 |
> | BLINK | ALL | 70.9 | 64.2 | 52.1 |
> | INST-IT IMAGE | MC | 84.9 | 77.3 | 86.7 |
> | INST-IT IMAGE | OE | 78.0 | 69.3 | 74.3 |
> | INST-IT VIDEO | MC | 65.5 | 52.9 | 71.3 |
> | INST-IT VIDEO | OE | 57.9 | 54.8 | 40.6 |
>
> **W6 Writing**. Thanks for your carefulness and attention to these details that improve our paper. For your questions,
> **(a) Spelling mistakes**. We will correct them in the updated version.
> **(b) Captions**. Please refer to https://anonymous.4open.science/r/csteer-2119/Reb2_Fig3_vector.png and https://anonymous.4open.science/r/csteer-2119/Reb3_Fig5_data.png, where we include detailed captions for them for clarity. For mentioned tables, we will update them in the updated manuscript.
> **(c) Figure 3**. We kindly point out that Figure 3 illustrates the Rewrite v.s. Rollout design for building contextual vectors. Please refer to the aforementioned https://anonymous.4open.science/r/csteer-2119/Reb2_Fig3_vector.png for details.
> **(d) SoM**. It is short for Set-of-Mark prompting, a method from [Yang et al.,2023].
>
> **Q1 Layer**. Yes. The layer sweeps refer to applying vectors on layers one by one, to observe which is effective. We observe medium-to-late layers are the most consistently effective ones. For our experiments, we apply contextual vectors to layers 20 ~ 24 for Qwen3-VL and 24 ~ 28 for InternVL-3.5.
>
> **Q2 Why Minimal Impact**. We believe these results highlight the importance of marker tokens. To further ablate, we apply contextual vectors to non-marker tokens. We kindly highlight that this result cannot be on par with that of steering all and marker tokens, suggesting that non-markers play a less important role.
>
> | Setting | GAR |
> |---|---:|
> | Qwen3-VL | 58.9 |
> | on all | 65.4 |
> | on marker | 66.4 |
> | on non marker (new) | 59.6 |

---

> > ### Author Rebuttal · Reviewer_mxzG · 2026-04-02
> >
> > **W1.** The authors mention that they do not average the vectors for the different criteria. But that raises more severe concerns. Each criterion targets a specific behavior in the model and captures the latent factors required to steer the model in that direction. However, practically, there can be combinations of those criteria, and using them individually does not seem to be a good approach. Moreover, there is no way we can know beforehand whether the response will use a specific criteria vector for steering, because we do not know what the issue is with that response (for example, whether it does not refer to the region or what the specific error is).
> >
> > The authors have addressed some of my other concerns, but I am still leaning towards rejecting this paper.

---

> > > ### Author Response · Authors · 2026-04-03
> > >
> > > We thank the reviewer for the feedback. We would like to raise up two questions regarding replies from the reviewer.
> > >
> > > **First**, apart from **w1**, especially we cost more than $400 on **w5** for testing proprietary models across multiple datasets, have our responses addressed all other **seven** concerns the reviewer suggested? .
> > >
> > > **Second**, we respecfully point out that the reviewer regarding **w1 averaging** is **self-contradictory**.
> > >
> > > In the initial review, the reviewer points out that averaging directions are not reasonable.
> > >
> > > >  These two vectors obtained from RvNR and EMvMS methods are in two different directions. So, taking an average of these does not guarantee the necessary multi-region referring behavior.
> > >
> > > In the rebuttal acknowledgement, the reviewer think averaging directions are practical.
> > >
> > > > Each criterion targets a specific behavior in the model and captures the latent factors required to steer the model in that direction. However, practically, there can be combinations of those criteria, and using them individually does not seem to be a good approach.
> > >
> > > Again, they are design variants (as mentioned by reviewer **Lyu8**), for the purpose of comparisons. We point out that none of the design variants was proposed by previous papers.
> > >
> > > We believe merging vectors (proposed by the reviewer) is a potential approach. But we regretfully point out that it is  **irresponsible** for the reviewer to **hold a unclear, self-contradictory position** across the phase. This behavior should not be encouraged, considering ALL readers who have put their efforts, including authors, other reviewers, the area chair, and finally the ICML proceedings.

---

### Official Review · Reviewer_sKs7 · 2026-03-13

**Soundness:** 3
**Presentation:** 3
**Significance:** 3
**Originality:** 3
**Overall Recommendation:** 5
**Confidence:** 4

**Summary:**

This paper studies the problem of multi-region visual referring in large multimodal models (LMMs). While recent multimodal models perform well on global visual understanding tasks, they often struggle when reasoning requires identifying and relating multiple localized regions within the same image. Existing solutions typically rely on task-specific architectures or require additional fine-tuning, thereby reducing flexibility. To address this limitation, the authors propose Contextual Latent Steering (CSteer), a training-free method that enables general-purpose LMMs to better handle multi-region referring without modifying the model architecture or retraining.

The key idea is to construct contextual steering vectors that represent correct multi-region reasoning behaviors. These vectors are obtained from contrastive pairs of positive and negative localized captions that represent correct and incorrect referring behaviors. During inference, the vectors are injected into the model’s hidden representations to steer the internal computation toward more reliable region-level reasoning. Experiments on multiple multi-region referring benchmarks show that applying CSteer improves performance of general-purpose LMMs and can even outperform specialized referring models that require dedicated architectures or supervised training.

**Compliance With Llm Reviewing Policy:**

Affirmed.

**Key Questions For Authors:**

1. The proposed contextual vectors are derived from contrastive examples consisting of positive and negative localized captions. If the contextual vectors were generated using different datasets or alternative caption generation strategies, would the resulting steering directions remain effective, or would performance vary significantly?

2. Have you evaluated the method on settings with a substantially larger number of visual prompts or more densely populated scenes, and if so, how does performance change in these cases?

3. Have you analyzed intermediate activations, attention distributions, or token-level reasoning patterns to verify that the model’s internal behavior is shifting toward more context-aware reasoning rather than simply altering the final prediction?

4. Do you expect the contextual vectors learned for one model to transfer to other architectures, or must new steering vectors be constructed for each model independently?

5. Could you provide further analysis on how the performance varies with different steering magnitudes or when the intervention is applied at different layers of the model?

**Limitations:**

yes

**Strengths And Weaknesses:**

While many multimodal systems demonstrate strong performance on global scene understanding or single-object localization, their ability to reason about multiple interacting regions within the same image remains limited. The authors clearly motivate this problem and show that such scenarios require models to both differentiate between multiple regions and incorporate global contextual reasoning. The proposed approach is appealing because it attempts to solve this problem without introducing additional training or architectural complexity, instead relying on representation-level intervention during inference.

From a methodological perspective, the core idea of contextual latent steering is conceptually straightforward and technically sound. The approach builds on recent work in representation steering and activation editing, and adapts these ideas to the multimodal setting. The use of contrastive pairs consisting of correct and incorrect localized captions provides a reasonable mechanism for identifying latent directions associated with improved multi-region reasoning. The method of constructing contextual vectors as differences between hidden states of positive and negative examples follows a standard formulation in representation editing literature, and the authors clearly describe how these vectors are injected into the model during inference. The technical formulation appears coherent and consistent with existing approaches to representation-level intervention.
The paper tests the approach on several benchmarks that explicitly require multi-region reasoning, including tasks that involve spatial relationships, counting, and contextual descriptions. The results indicate that general-purpose LMMs enhanced with CSteer can outperform specialized referring models that require task-specific training or additional region encoders. This is a meaningful finding because it suggests that some limitations of multimodal models may be due to how their representations are utilized rather than deficiencies in the base model architecture itself.

The paper is generally well structured and the narrative is easy to follow. The authors also do a good job of positioning their work relative to existing approaches that rely on fine-tuning or specialized region encoders. The conceptual framing of multi-region referring as a latent capability that can be activated through representation steering is particularly clear and helps differentiate the paper from purely architectural approaches.

Despite these strengths, there are several aspects of the work that could be further clarified or analysed. The experiments focus primarily on specific multimodal models and benchmark datasets, and it remains somewhat unclear whether the same steering vectors would transfer effectively across different model architectures or across tasks with substantially different visual reasoning requirements.
Another aspect worth exploring further is the stability of the steering mechanism across different levels of intervention. The current experiments demonstrate that applying contextual vectors improves performance on the evaluated tasks, but the paper does not fully examine how sensitive the results are to the magnitude of the steering signal or to the choice of layers where the intervention is applied. A more systematic analysis of how steering strength affects model behavior could provide additional insight into the mechanism underlying the improvements.
It would be nicer to include additional diagnostic experiments that examine how the steering process affects internal representations or attention patterns within the model. Since the paper’s central claim is that contextual vectors modify internal reasoning behavior rather than simply altering output predictions, further analysis of representation changes could help validate this interpretation. For example, visualizing how attention over regions changes before and after steering or analyzing intermediate reasoning steps could provide stronger evidence that the model is genuinely shifting toward more context-aware reasoning.
Finally, while the results show improvements over specialized referring models, it would be useful to include experiments that test the method on more complex scenes containing larger numbers of regions or more ambiguous spatial relationships. Multi-region reasoning becomes substantially more challenging as the number of interacting entities increases, and evaluating the method under these conditions would provide a clearer picture of its scalability.

---

> ### Author Rebuttal · Authors · 2026-03-31
>
> We thank the reviewer for the recognition of our motivation, simplicity of our design, and our difference against other visual referring approaches. Please see our point-to-point responses to the reviewer questions section.
> **Q1 Data for Captioning**. As mentioned in Appendix A.1, we use INST-IT as data for building contrastive examples and contextual vectors. To verify this, we use multiple data sources additionally (including osprey and mdvp-data, from PixelRefer [Yuan et al., 2025]) to build contextual vectors, while others remain the same as pipeline in our paper. We apply these vectors to Qwen3-VL and test on GARBench [Wang et al, 2025b]. The results are as follows.
>
> | Vector Source | GAR MC | GAR OE (DET) |
> |---|---:|---:|
> | baseline | 63.9 | 58.9 |
> | osprey (new) | 65.1 | 63.6 |
> | mdvp (new) | 65.6 | 64.5 |
> | inst-it | 65.8 | 66.4 |
>
> **Q2 Crowded Scene**. We kindly point out that INST-IT data we evaluate contains densely populated, instance-level scenes (15 objects maximally). CSteer achieves 1.1% gains over Qwen3-VL on average.
>
> **Q3 Attention**. Thanks for the constructive feedback. We have analyzed how relative attention shifts after applying CSteer for samples in Figure 2 (second row). Visualizations are given at https://anonymous.4open.science/r/csteer-2119/Reb1_attention.png (we use a different color palette to present **changes** on relative attention). According to the results, CSteer enables LMMs to be more context-aware.
>
> **Q4 Cross Models**. We test whether contextual vectors from Qwen3-VL improves InternVL-3.5 for contextual referring. The results are as follows. Surprisingly, we observe that vectors from Qwen3-VL could guide InternVL-3.5 to some extent. This could be attributed to the same LLM they use (Qwen3-8B-Instruct).
>
> | Test Model | Vector Source | GAR MC | GAR OE (DET) |
> |---|---|---:|---:|
> | InternVL-3.5 | - | 50.9 | 49.5 |
> | InternVL-3.5 | Qwen3-VL | 53.1 | 55.1 |
> | Qwen3-VL | Qwen3-VL | 51.4 | 53.2 |
>
> **Q5 Magnitude and Layer**.
> **(a) Magnitude**. We set steering magnitude to 1.0 across evaluation setups. Here we apply three more magnitudes (0.25/0.5/2.0) and test how Qwen3-VL could be improved on GAR-Bench. Results for multiple-choice questions (MC) and two open-ended generations (OE) are in the table below.
>
> | Qwen3-VL | GAR MC | GAR OE (DET) |
> |---|---:|---:|
> | 0.00 (bas) | 63.9 | 58.9 |
> | 0.25 (new) | 64.6 | 64.5 |
> | 0.50 (new) | 64.9 | 64.5 |
> | 1.00 (ours) | 65.8 | 66.4 |
> | 2.00 (new) | 65.5 | 68.2 |
>
> **(b) Layer**. We refer the reviewer to Figure 5, where we present results when intervention is applied to different layers of the model. Note that CSteer takes effect mainly in medium-to-late layers. For our experiments, we apply contextual vectors to layers 20 ~ 24 for Qwen3-VL and 24 ~ 28 for InternVL-3.5.

---

> > ### Author Rebuttal · Reviewer_sKs7 · 2026-04-02
> >
> > I select this option as some concerns (e.g., synthetic data reliance and limited novelty) are not fully resolved and require more than a short rebuttal. However, they do not affect my overall positive assessment.

---

> > > ### Author Response · Authors · 2026-04-02
> > >
> > > Thanks for the overall positive assessment of our study from the reviwer, as well as the suggestions for improving our paper. We hope our new experiments have addressed some concerns. We would be happy to further address remaining questions if there is any.

---

### Official Review · Reviewer_Lyu8 · 2026-03-18

**Soundness:** 3
**Presentation:** 2
**Significance:** 3
**Originality:** 3
**Overall Recommendation:** 4
**Confidence:** 4

**Summary:**

This paper builds upon CAA (Contrastive Activation Addition) to steer MLLMs during multi-region visual referring inference without any instruction-tuning or RL training. At its core is how to model the contrastive direction used to steer MLLMs, i.e., the contrastive signal between positive and negative pairs determines the steering direction and ultimately model performance. The paper is well-motivated with detailed ablations, and the proposed CSteer method improves upon the set-of-marker approach.

**Compliance With Llm Reviewing Policy:**

Affirmed.

**Key Questions For Authors:**

Authors should add discussions on efficiency.

**Limitations:**

yes.

**Strengths And Weaknesses:**

Strengths:

1) The paper is well-motivated, given that current MLLMs fails at multi-object visual referring, i.e., Fig. 2 in paper, involving the SoM approach with context steering is a solid improvement.
2) The performance is improved on multiple benchmarks, the devised variants (Eq. 5-8) of contrastive signals are intuitive, and the conducted ablations support paper's claims.

Weaknesses:
1) There is no efficiency comparison, since CSteer requires model to generate 8 rollouts for each query, and leverage two extra LLMs before generating the final response, i.e., a text-only LLM (Qwen-2.5-72B) to judge the score of each rollout given the ground-truth answer instance captions or answers, and a LLM (GPT-4o) to rewrite wrong rollouts while keeping original response's style and structure, I believe there are extra costs but paper misses any discussion on this respect.
2) The writing of section 3 method could be improved, i.e., clearly define x_{+} is ground-truth caption or answer, and define f is actually teacher forcing by inputting the concatenated v, p, and x_{+/-} together into MLLM to directly extract its hidden representation. Clearly demonstrating this could ease the burden of potential readers.

---

> ### Author Rebuttal · Authors · 2026-03-31
>
> We thank the reviewer for the recognition of our work, and constructive feedback. Please find our replies as follows.
> **W1 Efficiency**. We present the wall-clock time of extra inference stages below (per sample). This includes rollout (8 rollouts for each query), scoring (with Qwen2.5-72B) and rewriting (with Qwen2.5-72B). We use 4 RTX 3090s to implement the aforementioned stages. This could be further accelerated with quantized judges (such as Qwen2.5-72B-AWQ) or better GPUs. We will include this in our manuscript for reference. Note that these vectors are only built once for all images we test.
>
> | rollout | scoring | rewriting |
> |---:|---:|---:|
> | 5.256 (s) | 3.785 (s) | 7.655 (s) |
>
> **W2 Writing**. Thanks the reviewer for pointing this out. As the reviewer understands, the contextual vectors are extracted by teacher forcing, via inputting the concatenated v, p, and x_{+/-}. We will revise the method section as suggested.

---

### Decision · Program_Chairs · 2026-04-30

**Decision:**

Accept (regular)

**Comment:**

This paper proposes CSteer, a training-free inference-time steering method for improving multi-region visual referring in large multimodal models. The approach is well motivated and practically relevant, addressing a clear limitation of current systems. It is simple to integrate, does not require fine-tuning, and demonstrates consistent empirical gains across multiple benchmarks. The experimental results and ablations generally support the effectiveness of the method, and the work aligns well with the emerging paradigm of inference-time control over frozen foundation models.

At the same time, several reviewers raised concerns regarding clarity of notation and presentation, lack of detailed efficiency analysis, limited discussion on transferability and scalability, and insufficient positioning with respect to prior steering methods. Some baseline comparisons are also missing or not fully aligned. While these issues should be addressed in revision, they do not fundamentally undermine the contribution. Overall, I assign a weak accept (low priority: accept if there is room in the program), as the paper offers a useful and timely idea but would benefit from improved clarity and more comprehensive analysis.